# "You see this thing is hard. . . ey, this thing is painful": The burden of the provider role and construction of masculinities amongst Black male mineworkers in Marikana, South Africa

Yandisa Sikweyiya[1,2]*, Sebenzile Nkosi[3], Malose Langa[4], Don Operario[5], Mark N. Lurie[6]

**1** Gender and Health Research Unit, South African Medical Research Council, Pretoria, South Africa, **2** School of Public Health, Faculty of Health Sciences, University of the Witwatersrand, Johannesburg, South Africa, **3** Alcohol, Tobacco, and Other Drug Research Unit, South African Medical Research Council, Pretoria, South Africa, **4** School of Community and Human Development, University of the Witwatersrand, Johannesburg, South Africa, **5** Department of Behavioral and Social Sciences, Brown University School of Public Health, Providence, RI, United States of America, **6** Department of Epidemiology, Brown University School of Public Health, Providence, RI, United States of America

* yandisa.sikweyiya@mrc.ac.za

**Data Availability Statement:** All relevant data are within the manuscript and its Supporting Information files.

## Abstract

In this paper we examine men's insights on how migration and minework affect their perceptions and performances of masculinity in the settings of minework and in their "real home" communities and explore the potential consequences of masculinity constructions for their own and their family's health. This study used qualitative methodology. Findings are based on 13 in-depth interviews conducted over two phases of data collection with adult men who were either working or seeking work in the mines in North-West province, South Africa. Data suggest that for these men, migration to and working in the mines meant they must straddle the temporal space of work and the rural home space. For these men, the role of provider was an inescapable demand and, resulting from migration for work, their experience of fatherhood was solely centred on material provision with little or no emotional involvement with their children. Findings further illustrate the impact of minework on men's health and livelihoods-resulting in some men reimagining and seeking to create alternative career paths for their children. There is pressing need for labour reforms on the employment conditions of low-paid mine workers to enable them to reinforce their livelihoods and secure better futures for their families. Gender-transformative interventions which aim to transform ideas of masculinity that emphasize providing rather than emotional involvement with children are also needed.

## Introduction

Historically, the discovery of gold and other minerals in South Africa led to a booming mining industry that continues to make a significant contribution to the country's gross domestic product. At the same time, however, the mining industry has sustained racialized capitalism–

**Funding:** This work was supported by the National Institute of Mental Health and the South African Medical Research Council (R01 MH106600) as well as NIH Fogarty Grant (D43TW011308-01. The content is solely the responsibility of the authors and does not necessarily represent the official views of the National Institutes of Health or the South African Medical Research Council.

**Competing interests:** The authors have declared that no competing interests exist.

i.e., the broader global phenomenon of economic disenfranchisement of Black people which has its roots in colonial systems that provide economic opportunities for white people while depriving these to other groups [1]. The South African mining industry grew its profits from the cheap labour provided by thousands of Black men who were recruited from the homelands and neighboring countries during apartheid [2]. Previous research has described health vulnerabilities among Black South African men generally, as well as among Black South African men working in the mining industry. Building on social and anthropological research by Hunter [3, 4] and others [5–8], this paper centers the analysis on intersections between notions of masculinity, work, and economic and familial responsibilities among Black South African mine workers, and explores implications of these intersections for health, wellness, and gender-transformative interventions.

Scholarly work on men and masculinities has increasingly demonstrated that gender identities are intertwined with racial and socio-economic politics that have resulted in the socio-economic marginalization of Black men and women [9, 10]. Such theorization about marginalized masculinities is often juxtaposed against Connell's theory of masculinities [11]. Connell recognizes patriarchal privilege as an organizing principle that confers men with gender-based advantages relative to women, but also contends that in any context there are multiple constructions of masculinities–each of which might uniquely hold unequal relationships to control over women, unequal access to resources and opportunities, and afforded unequal regard by other men. Hegemonic masculinity is both a highly popular and contested concept in the scholarship of men and masculinities [12], particularly in the global South [13]. The contestation has centered around the position of Black men in society and the role of power as the main defining property of hegemony. Given Black men's disenfranchised material conditions, both globally and in South Africa specifically, the concept is highly problematic for understanding the lives of Black men [14]. For example, Ratele [14] calls for a view of hegemonic masculinities among Black men as hegemony in marginality, giving rise to a recognition that Black masculinities are constructed within contexts of patriarchal privileges that are tied to maleness yet also shaped profoundly by race- and economic-based marginalization.

Some scholars have argued about 'masculinity in crisis', which involves recognition of the structural conditions that destabilize assumptions of hegemonic masculinity and contribute to men's failure to fulfill masculine ideals [3, 6, 15, 16]. Across many cultures, men are ascribed the roles of head of households, protectors and providers [17, 18]. The current South African social and economic conditions characterized by high levels of unemployment or precarious job opportunities make it difficult for many men, particularly Black men, to achieve local forms of hegemonic masculinity, including securing jobs, marrying, fathering children, and establishing their own households [19]. They often cannot afford to pay *ilobolo* (bridewealth), and *inhlawulo* (a fine for impregnation which grants the paternal rights and access to the child) [15, 20, 21], or build their own *umuzi* or *motse* (homestead or house) [19]. The lack of resources makes it difficult for many Black men to establish families or establish themselves and take on the role of legitimate fathers particularly when they father children outside the institution of legalized marriage [22]. Furthermore, many men are unable to fulfill their traditional roles as 'providers' and 'breadwinners' to support their families. Hunter [3] describes these men as men without *amandla* (power).

Mining remains a very significant economic activity in South Africa–both for the national economy as well as for mineworkers and the families that they support. Poor Black African men from rural areas of South Africa continue to disproportionately provide workforce for the mining sector [23]. In addition to being historical sites for the economic exploitation of Black men's labor, mining locations are also settings for workers' increased vulnerability to occupational and social-related health risks such as respiratory illness [24, 25], infectious diseases

including HIV [26–29], violence [2, 30], and mental health problems [31, 32]. Mining locations are also potent sites for social unrest and active protest among workers. For example, in August 2012, a mass protest of about 3000 mineworkers broke out in the Platinum mines in Marikana, North-West province. This was arguably the longest strike in the mining sector since the advent of South Africa's democracy in 1994, lasting over 18 weeks [33]. Amongst other demands, the mineworkers were seeking better living and working conditions and a livable wage of R12000 per month. On 16 August 2012, South African police opened fire on striking miners using automatic weapons, killing thirty-four [34].

Mining sites provide a compelling backdrop to examine the intersections of marginalized masculinities, occupational setting, and health vulnerability of Black South African male workers. The specific aim of this qualitative study was to explore male mineworkers' insights on the ascribed "provider" role and how it reflects the construction of masculinity(ies) and relates to health consequences in the specific context of minework in Marikana, North-West province, South Africa.

## Material and methods

### Study design

This was an explorative qualitative study using in-depth interviews (IDIs). We adopted this study design because we were interested in gathering narrative insights into the lived experiences of Black African male mineworkers with regards to the provider role and how it is linked to the construction of masculinity(ies) in this setting.

### Study area

Participants lived in Nkaneng and Wonderkop settlements located outside the premises of the Lonmin mines in Marikana, North-West province, South Africa. While these two communities were mainly comprised of informal houses, there were several houses built of cement blocks present in both settlements. These communities provide residence primarily to men who are either working or seeking employment in the platinum mines. Mineworkers in this setting have the option of residing in company-provided single sex housing or family units; alternatively, they can receive a housing allowance and seek accommodation outside of the mine premises.

Empirical evidence from South Africa shows that informal settlements are characterized by poverty, poor infrastructure and hygiene, high unemployment rate, with studies suggesting a strong association between these conditions and a range of serious public health problems including interpersonal violence [35, 36] and HIV and AIDS [37, 38].

The majority of participants resided in Nkaneng, next to the Koppie-the specific site of police violence during the 2012 strike [39]. Many participants lived in small shacks or rented "back rooms" with no access to basic services such as running water, electricity and hygienic drainage systems. The streets were not tarred. These issues were raised during the strike–especially the lack of decent houses for mine workers [39]–but little has changed materially since.

### Study participants

Table 1 provides descriptive sample characteristics. Most men in our sample were either currently working, seeking employment or were recently employed on the mines at the time of the study. Men's ages ranged between 28 and 63 years. In terms of formal education, levels of completed education ranged from grade four (i.e., primary/elementary education) to grade 12 (i.e., high school diploma). None of the men interviewed had a tertiary education. Most were

**Table 1. Characteristics of participants.**

| Pseudonym | Age | Relationship status | Grade passed | Ethnic group |
|---|---|---|---|---|
| Boitumelo | 33 | Married (traditionally) | 9 | Sotho |
| Luzuko | 28 | Married (traditionally) | 8 | Xhosa |
| Mandla | 53 | Married (traditionally) | 9 | Xhosa |
| Wiseman | 41 | Married (traditionally & civil) | 9 | Xhosa |
| Ntsizwa | 35 | Married (traditionally) | 4 | Pondo |
| Vusumzi | 48 | Married (traditionally) | 6 | Mpondomise |
| Vuyani | 52 | Married (traditionally) | 7 | Xhosa |
| Liwa | 52 | Married (civil) | 12 | Xhosa |
| Sinovuyo | 31 | Married (traditionally) | 12 | Xhosa |
| Buhlebezwe | 47 | Married (traditionally) | 7 | Xhosa |
| Oupa | Pensioner | Married (traditionally) | 9 | Tswana |
| Morena | - | Single (cohabiting) | - | Tswana |
| Lethabo | 63 | Single | 8 | Tswana |

married, with the majority married traditionally, which is legally recognized in South Africa under the Customary Marriages Act, 1998. Amongst those married, almost all reported that their wives and children were permanently residing in their rural homes in other provinces. In terms of ethnicity, most participants (N = 7) identified themselves as Xhosas, one as a Pondo, another as a Mpondomise, one a Sotho, and three as Tswanas. All men reported to have had at least one biological child.

## Data collection

The data came from 13 in-depth interviews (IDIs) conducted over two phases of data collection. The first phase of data collection was done in Nkaneng in 2016, where nine IDIs were conducted. This was followed by a preliminary data analysis which enabled us to identify new directions and emerging patterns in the data to pursue in subsequent interviews [40, 41]. We then revised our interview guide and conducted a further four IDIs in Wonderkop in June 2017.

For the first phase of data collection, a purposive sampling technique [40, 42] was used to select nine men who resided in Nkaneng and had been living there for a period of five or more uninterrupted years-a time period which overlapped with the occurrence of the 2012 mineworkers' strike. The first author spent approximately 3 months visiting homes in Nkaneng and inviting men who met the study criteria to participate in the study. The Marikana mineworkers' strike was a very painful and traumatic experience which left a legacy of hurt, anger and mistrust amongst people [see 39]. Therefore, concerned with their safety, four men expressed discomfort with the research focus and declined the invitation to be interviewed.

For the second phase of data collection, we used a theoretical sampling technique [40, 41] to identify four men who were receiving professional counselling to deal with the effects of the 2012 mineworkers strike. Recruitment was facilitated by lay counselors working with mineworkers in Marikana, who provided referrals to the study.

Almost all participants had actively participated in the strike. In the interviews, participants were asked to share their views on what they think the roles of men are within families, what challenges they had personally experienced in fulfilling expectations associated with their role as men in their families, what working in the mines mean for them as a man, what are the most important things to them as a man, what they need to achieve as a man for them to view themselves as a successful man, how much would they say they have achieved as a man, what

kind of a man one needs to be to fit well among other men in the mines, what kind of men are not popular or considered weak men in the mines. Next, men were asked about the 2012 mineworkers' strike, their views, involvement and experiences of the strike. The IDIs allowed the intricacies and contradictions of real lives and personal stories to emerge, without men feeling obliged to represent themselves in a social desirable manner, as they would likely do when in a group [43]. IDIs lasted approximately 1 hour, ranging from 30 minutes to 1.5 hours. All IDIs were digitally-recorded and conducted by the first author using a mixture of isiXhosa, isiZulu and bit of Setswana–and the participants could express themselves through one or a mix of these languages. Digitally recorded IDIs were transcribed verbatim and translated into English by a professional translation and transcription company. Upon completion of the transcription, recordings were checked to ensure transcription accuracy.

## Data processing and analysis

We analysed data inductively employing a thematic analysis approach [40, 44]. All authors were involved in various stages of data analysis. We first read and re-read the transcripts individually to familiarize ourselves with the content of the transcripts. Next, we established some general codes which somewhat resembled the questions in the interview guide. Further to this, text which seemed to fit together was grouped under a specific code [40]. Next, we examined the data and found numerous open codes. Open codes which were deemed to be similar were clustered together under clearly defined themes. Thereafter, we explored the associations between the themes and interpreted what we saw emerging. Last, we compared our findings with those of similar published studies [40, 41].

## Ethical considerations

This study was approved by the South African Medical Research Council Human Research Ethics Committee (clearance no. EC027-8-2015). Before conducting the IDIs, the first author reviewed the participant information and consent form with the participants. The discussion included the rights of participants in the study, and risks and benefits of participating in the study. Written informed consent was obtained from all participants. We have replaced the names of study participants and two locations in which the study was conducted with pseudonyms to protect the identity of the participants. Participants were reimbursed with R150 (approximately $10) for their time.

## Results

Five main themes emerged during data analysis. First, participants' narratives depicted the dynamic construction of masculinities as mineworkers straddled both the temporary space of work and the "real" rural home space. Second, economic provision was characterized as an inescapable demand on poor Black male mineworkers, imposed by society but internalized as an intrinsic part of their manhood. Third, despite economic hardships and instability, participants consistently endorsed the fundamental societal expectation for men to provide for their children. Although men conceded that attempting to achieve the provider role was painful, they did not contest this demand on them. Fourth, narrative accounts revealed impacts of the intersection between masculinity and minework on psychological and physical health. Last, we show that the men's efforts to provide economically for their children and family continued, despite their concessions that they had limited chances of succeeding. With this realization, some men sought to create alternatives (e.g., investing in education for their children) to ensure the future survival of their families.

## 'The land that we live in here is not ours': Constructions of masculinity straddling spaces

Men interviewed spoke of the responsibility of straddling the rural home and work spaces and negotiating the masculinity-related expectations that were specific to each space. The mining context where the men worked and resided was viewed as a temporary space for the purpose of generating income and subsequently transferring income to the 'real home' space to provide for their families. This is illustrated in Vusumzi's narrative below:

Here in Marikana it is just a place of work. Where I am supposed to do something is at [rural] home. I am working here so that they will not suffer back at home. (Vusumzi, aged 48)

A similar sentiment was shared by Liwa:

I do have another house back at [rural] home, I do not want to lie but I am here for work purposes. I am here because I thought about the future of my wife, kids and myself so I am not here forever, I came here for work purposes. The land that we live in here is not ours however we stay because we want to support our families. (Liwa, aged 52)

In the 'real home' space, ideal masculinity was defined by establishing a family, being a provider, and having certain possessions, like livestock [45], all of which bestowed upon men a desirable masculine identity and respectable social standing among other men in the home community. Wiseman's quote below clearly summarizes this notion of a desirable masculine identity:

For you to be an honored man you must be married. A man that is married is always followed by children. Then when the children are there, they will call you father, which means they are scared of you and you also need to set an example and all that makes you as a man feel honored. You become an example to them [children] and they can easily say 'there is our father' and not call you by other names. You must also have livestock in your yard like chickens, cows, sheep you name them, so people can see this is a home belonging to a man. Having livestock also puts you in a position to say something in meetings amongst other man, because you are also a man now (laughing) . . . Your voice must be heard, and you should appear to be a man working towards something that can be noticed. (Wiseman, aged 41)

In his interview, Morena explained that a man who fails to achieve certain masculinity milestones in the rural home space will not be recognized as a man by other members of the community:

Morena: When you are a man you are supposed to get married, build your own house and have your own family.

Interviewer: If you have not done that how does the community perceive you?

Morena: Ey, you are not recognized. . . you know they do not consider you as a man, they undermine you. (Morena, age unknown)

As reflected in the extracts above, the men's definition of manhood in the rural home space emphasize leadership, being accomplished, establishing and providing for a family–aspects of masculinity that have been associated with traditional patriarchy [46, 47].

In the temporary space of work, in which men were deprived of external symbols corresponding to ideal masculinity (like having a family and livestock) that are present in the 'real home' context, participants described a reconfigured masculinity. This reconfigured masculinity was constructed through the demonstration of monetary prowess and behavioral performances strikingly akin to the concept of hyper-masculinity [48]. For example, in these temporary work spaces, men displayed their reconfigured masculinity by spending money publicly (e.g. in bars/taverns), having sexual relationships with multiple women, drinking alcohol, and similar performative gestures that gained admiration and recognition by women and other men in mining environments.

> Some men like nice times, they forget where they are coming from and where they are going... They 'groove', they call it 'groove' (having parties with friends). (Boitumelo, aged 33)

Participants described that transactional sexual relationships in Marikana were common, and typically were initiated based on men buying alcohol for women whom they met at bars/taverns. While these relationships often involved once-off sex and money exchange, some of these women took on the temporary role of homemakers; the men, in turn, contributed to the material needs of these temporary partners and sometimes to the needs of her family.

Interviews revealed that some of these relationships developed into longer-term relationships. Hence, they were perceived and served as pseudo marriages, and provided men an avenue to enact a provider role status in the temporary work space.

> ...if she [non-marital sexual partner] is here (in the morning) she will want to stay behind and sleep and when you return from work, she has washed all the dishes, your clothes, and cleaned the house too. (Vusumzi, aged 48)

> A lot of men are staying with them [non-marital sexual partner] now like a wife and she cooks for him, does everything. A lot of men do that, they stay with women in their houses it's like she is his wife. Men do that here, a lot of men. (Mandla, aged 53)

The interviews suggested a contestation between the ideal masculinity which was hegemonic in the 'rural home' space and the hyper-masculinity in the temporary space of work. Disapproval was directed toward men who engaged in transactional relationships that benefited masculinity in the temporary space at the expense of providing financially for the man's family in their rural homes. These types of hyper-masculinity performances were deemed reckless and violations of the very reason for minework–i.e., prioritization of family provision.

> We [men] view him as a stupid man. There's a lot of men here that are like that. They finish their money here and don't send anything home. We see their wives coming here to collect the money. (Luzuko, aged 28)

In his interview, Lethabo expressed skepticism about sexual interactions and relationships between men and women that occur in the temporary work space–intimating that these exchanges in the temporary work space are centered on physical and monetary exchange, rather than emotion and partnership. In his own words:

> Woman are no longer in love with a man, they sell [sex], if you ask for love they do not do that, but you can get that by luck, but these days no, they are many and they want you to buy, you give her money and she gives you her thing [sex] and that's the end, then you go

in separate ways, they are not up to this thing of loving each other, staying together, you can get one but they are no more. (Lethabo, aged 63)

Most of the men who were critical of their peers who spent money on women and in bars/taverns spoke proudly about taking on chores like cleaning their own rooms/houses, washing their own clothes, and cooking for themselves–activities that are traditionally aligned with the role of homemaker, and ordinarily a woman's role in patriarchal contexts. These men spoke of their engagement in domestic chores with great comfort and at times pride, particularly when they compared themselves to men who opted for temporary female partners to do these domestic chores for them. This is best illustrated in Vusumzi's account below:

Not that I would meet someone, and say come and cook for me [. . .] Even when I was in Durban, I had no problem with cooking when I was working in the firms. We are at the mines here, so most men cook for themselves. (Vusumzi, aged 48)

While these men took on the domestic chores in the temporary space of work, it is difficult to conclude that they were intentionally challenging norms of patriarchy and ideal masculinity, which is hegemonic in many Black communities in South Africa [13, 49]. Men viewed this as a temporary measure and roles from which they disengaged when their wives visited the work space or when they returned to their rural home space.

## Providing is an inescapable demand

Most of the men interviewed were committed to fulfilling the provider role, however, they also expressed views about the burdens associated with the role. For example, Luzuko's quote below acknowledges the taxing demands of the provider role:

I'm the man everyone is looking up to and depending on, so even if I complain I can't complain much because there is no one else to take these responsibilities. (Luzuko, aged 28)

The burden to provide appeared to have been particularly acute amongst men who were unemployed, as they were still expected to devise means to provide for their families in the rural areas. Buhlebezwe's extract below evidences this:

A man must [provide] support at home, yes. Money is expected from a man, so everything stops now if you are not working; nothing is progressing, you see. But because I had worked before and had cattle, when it is very difficult, I would say "no let's sell a cow because I don't have money." (Buhlebezwe, aged 47)

Narratives also revealed that the expectation for men to provide for their non-marital sexual partners also existed in the temporary space of work. When Oupa was asked how people in Marikana perceive a man who is not working, he posited:

Ey. . .you are nothing, no you are nothing, even the woman you are staying with, who looks after you here in Marikana if you [man] are not working you cannot take care of her, you should at least look for jobs. You see this thing is hard . . .ey, this thing is painful. (Oupa, a pensioner)

When asked what a man in Marikana should do to demonstrate that he is a man in his house, Oupa, a retired mineworker who said he was struggling to provide for his family,

contended that: "He must work, you see, and respect his job. . . and know how to get successful. . .. without working there's nothing he can do . . .yes, never without working" (Oupa, a pensioner).

Participant narratives emphasized the man's core responsibility to devise a means for financially supporting his family. However, it was clear in the men's narratives that this role expectation carried strong emotions and feelings when a man is unable to fufill these expectations. Luzuko described how he felt in instances when he was unable to provide for his family's needs:

> You feel very sad [if can't provide]. It's not a good thing to fail. You tell them [family] you will send money next week, but the child is sick now. (Luzuko, aged 28)

Fulfilling the masculine expectation of the provider was both a source of pressure and also a measure of self-worth and manhood, as seen in Wiseman's account below:

> It made me feel bad especially when I got a call from home asking for money I did not have because now back home they know I got paid and I haven't given them anything because I don't have. I had to provide for the needs of my other lady [non-marital sexual partner] here, her rent and my needs. Now I get reports that my children need this and that, it makes [my] heart ache very much. (Wiseman, aged 41)

Being a provider was particularly overwhelming for men who provided for their immediate and extended families (e.g. siblings and/or siblings' children) in the rural home space, as well as for their non-marital sexual partner(s) in the temporary work space. This burden reflects the common notion that in a traditional Black African family, expectations of being an economic provider often extend beyond one's immediate family. Indeed, participants noted that a man who earns an income or is employed is expected to support the entire family system. Many participants consequently felt that their salaries as mineworkers were not enough to support their families. Luzuko and Oupa's narratives to follow are illustrative:

> It's painful to know that you are the only one working and the whole family depends on me. And the money we get here doesn't even get to R10 000.00. Only when we work overtime, then we get a little more than that. I rent here and the kids are at school [in the rural home space] and I must send money for food and clothes. (Luzuko, aged 28)

> No, it [pay] was a cent that time, I was not getting enough money, per day I was paid R7.00 I worked as an electrician in the mine, I was paid R7.00 per day, not per hour but per day . . . No, it was very small and was not even funny because it was me and my children [depending on it]. (Oupa, a pensioner).

## Fatherhood centered on provision

Although most participants were married and had children, and hence had attained important features of the ideal masculinity in black African communities [50], they saw their families infrequently due to migration to work at the mines. Consequently, most men's experiences of fatherhood focused predominantly on economic provision. While the men's sense of fatherhood chiefly centered on providing, their jobs were precarious, they were poorly paid, and some were unemployed, making it difficult to fulfill the provider role. Participants Oupa and Buhlebezwe elaborate on this experience:

. . .. you see these boys? They are my children, there's an older one, his name is Bongani, they [sons] have children also, the two of them, another is my grandchild. And Ndaba, he has a daughter who is here. . . I buy them pampers [disposable diapers] and I am still feeding them because they are not working . . . there are no jobs. (Oupa, a pensioner)

It pains me a lot, you see, when a child says "father I have a shortage with something at school", or it's something like rent and I would realize that it's not month end yet and concede that there is nothing I can do and just tell her "please persevere". Or at least I would talk to someone who is at [the rural] home and say, "hey man can you lend me R2000, we will talk about what I will give you at home". So, I get by through making plans; when a child is asking for a big amount of money, I quickly call someone. So, when that person gives me the money, I would sell him something at home. That is the kind of life I am living. If I did not have a livestock, it would have been very difficult. (Buhlebezwe, aged 47)

Men who were unable to meet the responsibility of providing for their children attracted criticism from peers who would remind them of the primary reason for minework: to earn an income to provide for one's family. Luzuko elucidated how one male colleague was rebuked for neglecting his family in the rural home:

They [colleagues at the mine] sat down with him [a man who does not send money home] and reminded him what he is here to do and is not supposed to spend money like that. They told him 'man we are sent here, we are not here to play, don't forget!'. . .We are not here to work for ourselves, we are working for our families. (Luzuko, aged 28)

Fulfilment of the provider role was highly scrutinized and socially policed by peer mineworkers. Rhetoric around being a "real man" centered around how men spent their money, with providing for families and children being one of the main duties to fulfil. Indeed, participants noted how they can observe whether their peers adequately fulfilled the provider role expectation based on their children's appearance:

You see Sir, when you're working, there must be a difference between you and the person not working. You can buy groceries and send money home, but if you are not working, you can't do that. But some men don't send money home and now their kids suffer, they walk bare feet to school and at home there is no food, but the man is working. That is not right. (Mandla, aged 53)

## Impact of work on men's health

Against the backdrop of masculine role expectations predicated on economic provision for one's children and family, participants pointed out the daily dangers they face in their jobs and the constant threat minework posed to their lives. Men like Sinovuyo highlighted negative health impacts of minework:

Working in the mines most of the time it is nice and at the same time not nice in this way, most of the time here we use explosives and they are affecting us health wise, so it is not nice in that way. Even in the hospitals that we have here you would see that people are not alright although they will say it is other diseases and you will find that it is because of these explosives most of the time because some people have [a] weak immune system. TB is highest here in the mines because of the things we are working with; it is not nice in that way

because you end up seeing that your life will be shortened because of your working conditions. (Sinovuyo, aged 31)

Sinovuyo's assertion regarding the dangers inherent in minework was corroborated by men like Vusumzi and Ntsizwa, who emphasized the difficult conditions they work under and risks associated with their work. In their own words:

The conditions that we work under are very difficult . . .for example you may come here next week to ask for Vusumzi, and they would tell you that he has passed away, the mine collapsed on him. (Vusumzi, aged 48)

We just survive through God's protection. Otherwise when you go to work you don't know if you will be coming back or not, because the rocks can fall [on us] whenever they feel like. (Ntsizwa, aged 35)

These accounts demonstrate participants' concerns that minework threatens their ability to work in the future and, even more so, their fears that work-related impairment or death could leave their children without a provider. As such, the health hazards associated with minework undermined the provider status and notions of ideal masculinity conferred by this work, and further introduced anxieties about both short-term and long-term financial security for the men interviewed.

## "He (son) must not end up like me and work in the mines": Creating alternatives

The majority of participants felt that opportunities for work outside of the mines were limited, as it was increasingly "getting difficult for men like us to secure jobs in the country". Most participants sought work at the mines at a young age because their own fathers had struggled with providing for the family. Intensified by the masculine role expectation to provide, participants described having to terminate their education and relocate to the mines in order to supplement or take over their fathers' provision for the family. This dynamic is explained in Boitumelo and Morena's narratives below:

He (father) was not working, it was very hard to continue with schooling that is why we ended only in standard 7 [grade 5] and did not continue with schooling. It was very difficult growing up. We were hustling for small jobs while growing up until today. (Boitumelo, aged 33)

Ey it was hard because I was supporting my siblings, my two brothers and I don't have parents, my mother passed away in 1983. Eh. . . so I had to drop out of school and go work so that I can support my brothers and push them to go to school, that is what made me to continue working [in the mines], now if I wasn't retired because of sickness I'd still be working now. (Morena, age uknown)

Most men in our study described a sense of entrapment to work in the mines due largely to their low levels of education. Participants like Vuyani bemoaned the cycle between education, limited opportunities outside of minework, and implications for their health and further opportunities. Vuyani explained:

Mines are not a good working environment, unfortunately because of my level of education this is the only place with my bread and butter, if you worked a long time in mines once you retire you do not live for long. We just work because we are in need. (Vuyani, aged 52)

Men like Morena shared accounts of retiring early at the mines because of ill health. He posited:

> I was sick in 2013, yes, I retired from work by medical board, eh. . .after the strike you see the 2012 strike, it started while I was still working, it took six months. So after six months it was January 2013 I got sick you see, I was suffering from TB and I found that the mine is releasing me, I was given a medical board, so even now I am not working. . . eish I am hardly surviving. . . (Morena, age unknown)

In light of the emotional and economic struggles associated with fulfilling the provider role through minework, participants imagined better opportunities and career paths for their children. They perceived education as a means to access better opportunities for employability and financial stability. Participants like Vusumzi and Luzuko thus encouraged their children to complete higher levels of education in order to enhance their employment prospects. They explained:

> I am also giving him [son] a bright future so that he will not end up like me and work in the mines. Yes, he must not end up like me and work in the mines. (Vusumzi, aged 48)

> I would love to send my children to tertiary [schooling], and they become lawyers. I would like to be that father that has children that are lawyers. My father didn't have children that were lawyers because we didn't finish school. (Luzuko, aged 28)

While all participants had internalized the provider role and did not contest this responsibility, many conceded that meeting the financial needs of their families is unattainable and their ability to provide for them is threatened by the precariousness of minework and job scarcity in South Africa. Men like Buhlebezwe were thus open to the possibility of women becoming providers and breadwinners in families, in the future. He posited:

> I'm telling myself that perhaps when the one [daughter] who is a girl who is at university can finish, it would not be the same. She can be able to contribute even with a R1000, as we are not yet at the stage of getting a pension, we are still young. So, whatever cent she would give us, it would make a difference. (Buhlebezwe, aged 47)

This example of readiness for women to become family providers and breadwinners suggests a window of opportunity for engaging men in gender-transformative work. Statements such as this reveal the capacity for these men to critically reflect on the impacts of provider role expectations on their health and consider alternative lives for their children.

## Discussion

In this paper we demonstrate how migration and minework informs the construction and performance of idealized masculinities in both the rural home space and the temporary space of work, and how these masculinities contest each other in the temporary space of work. Moreover, we present findings showing how poor Black African men who migrate for work from the rural areas to the mines in big cities must straddle these two spaces which prioritize distinct constructions and expectations about masculinity. There are implications emanating from this tension in the construction of masculinity for poor Black African men. First, they must use their meagre salaries to support themselves in the temporary space of work while also providing materially for their families and children who have remained in the rural home space.

Second, while away from their families in the rural home space, some men constructed and performed hypermasculinity aligning with expectations and opportunities in the temporary space of work. Men and masculinity studies have shown that this masculinity is performed through engagement in risky practices including having multiple sexual partners, harmful use of alcohol, and illicit drugs [48, 51]. Consistent with our findings, studies in resource-poor communities in South Africa and elsewhere suggest the performance of hypermasculinity and other exaggerated masculinities oftentimes leads to the detriment of the men who ascribe to role expectations [14, 49–51].

In this study we have shown that while minework enabled access to the ideal masculinities in both the temporary space of work and the rural home space, for most men, it did not enable a sustained ability for them to meaningfully provide for their families. In our sample, men who had recently retired or been retrenched from the mines reported struggling financially and failing to provide for themselves and their families. Thus, soon after leaving the mines, men and their families entered further into poverty. As Gibbs et al. [52] argue, livelihoods challenges for poor Black men in South Africa are 'embedded in a complex intersection of economic, political and social marginalization that they have experienced over many years and even inter-generationally, with histories of migration and distant fatherhood, first emerging on their grandfathers, continuing through to them' (p.9). Targeted research is needed to establish the long-lasting implications of migration and mine work for poor Black men in South Africa.

We have also shown that in the temporary space of work, men reconfigured their masculinity and constructed one which was predicated on public demonstrations of possessing money, which often manifested as frequent patronizing of bars and buying and drinking large amounts of alcohol. Often men also purchased alcohol for women in the bars which, in turn, facilitated transactional sexual relationships. The men's practices of reckless spending of money in bars, engaging in harmful consumption of alcohol, and transactional sexual relationships are consistent with the hypermasculinity that is hegemonic in resource-poor communities [53, 54]. While our findings support this literature, our analysis further revealed that some men questioned the hypermasculinity that circulated in the temporary space of work. These men's opposition to the hypermasculine position-which is predicated on economic provision, sexual conquest and transactional sex-opens a window for engaging these men in gender-transformative and HIV prevention work and assisting them to reflect on other positive forms of masculinity [54].

We have presented findings showing that men in our study internalized societal expectations to be providers for their families, and elevated the provider role to a key characteristic of idealized masculinities in both the rural home space and the temporary space of work [53]. Notably, endorsement and internalization of this belief persisted among this sample of men who expressed frustrations, anxiety and feelings of burden linked to the provider role. Published literature suggests that in many contexts in Africa, men are expected to provide economically for their families [6, 55]. Izugbara has argued that expectations that Black African men should be providers for their families prevail even in times of hardship. In his study, poor men in two Kenyan slums vehemently argued that a man cannot call himself a man if he fails to put food on his family's table [6]. Yet, in the current economic climate in South Africa which is characterized by high unemployment rates [56], many Black African men struggle to fulfill the provider role within their families [57, 58]. In line with this, Khunou (personal communication) contends that the current high rates of unemployment in South Africa suggest that the idea of men as sole providers is not sustainable. We concur with Khunou and further argue that there is a need for labour reforms to address employment conditions of low-paid mineworkers to enable them to strengthen their livelihoods and secure better futures for their families.

Our findings show that as most men in our sample had migrated to the mines for work, their experiences of fatherhood mainly focused on economic provision, with little involvement in nurturing and providing emotional care for their children. While literature on fatherhood in South Africa is increasingly showing that fatherhood is seen by some men as going beyond material provision to include nurturing and provision of emotional care to children [20, 59], our findings conversely show that fatherhood among migrant mineworkers continues to be tied to material provision. This finding can be explained by literature which suggests that this way of fatherhood emanates from both historical and current reality of many poor Black African men who have to travel long distances to work or seek employment in the mines and only visit their rural homes and children infrequently [2].

We have presented a finding demonstrating that men in our sample were low-paid mineworkers with precarious jobs, while others were unemployed, and how this made it difficult for these men to fulfill the provider role. Literature from South Africa suggests that the societal expectation for poor Black African men to meet the demands of being a provider has implications for the degree of pressure they may perceive or experience towards being unable to fulfill the provider role [3]. Poor Black men's inability to fulfill the provider role has been shown to have serious ramifications for their masculinity [20, 57, 58] and psychological health [58]. In light of this, Thomas and Krampe [60] underscore the need for gender-transformative interventions to support Black African men to shift their thinking about their roles as men–from one which amplifies the provider role to one that accentuates the importance of spending time with family and children, being available emotionally for their children and being a role model to them.

In this paper, we have demonstrated that some men perceived minework as dangerous and a threat to their lives, yet they felt trapped to continue working or seeking work in the mines as they had limited opportunities to secure jobs in other sectors because of their low levels of education. Prior qualitative research in resource-poor African communities has shown that poor Black African men aspire for improved livelihoods, enhanced social conditions and economic opportunities that would enable them to break the cycle of poverty for their children and families [6]. Indeed, some, men in our sample reimagined different lives for their children. These men envisioned a disparate future and career paths for their children-wishing to break the cycle of inter-generational migration for work to the mines. This finding suggests that within these men there were tensions in relation to performance of masculinities. Furthermore, there were tensions in relation to how these men felt compelled to be distant fathers separated from their families [15] and their appreciation and conviction that this arrangement was a prerequisite so that they can reimagine a better future for their children. However, based on this analysis, we contend that in trying to economically craft a better future for their children, these men were exposing themselves and their families to immediate hardships.

## Limitations

In this qualitative study, we interviewed men who were purposively selected because they had spent five or more years living uninterruptedly in Marikana and working or seeking work in the mines. Thus, our findings are not generalizable, yet we hope the insights we have collected are of interest to other similar African contexts. There is a possibility that some men may have felt obliged to respond in a particular manner, including feeling the necessity to describe themselves as 'real man' with traditional values or to represent themselves in a socially desirable manner with regards to their struggles for work and inability to provide for their families.

## Conclusions

We have shown that migration and minework allowed men to attain the dominant forms of masculinities that were salient in both the temporary space or work and the rural home space. Yet, we also demonstrated that migration and minework in this setting failed to provide poor Black African men with the ability to provide for their families in the long-term. Based on these findings, we argue that the provider role expectation placed heavy psychological and familial burdens on men by conferring ongoing pressures and anxiety to provide for their families and children, even within the current reality of job scarcity in South Africa and paltry salaries associated with minework. The men's critical appraisals of their own and their peers' masculine performances as well as their reimagining different lives for their children, both male and female, indicate an openness to disrupt their current constructions of masculinity. This finding suggests an opportunity for gender-transformative and parenting interventions which seek to transform notions of masculinity that essentialize providing instead of emotional involvement with families and nurturing of children.

## Supporting information

**S1 File.**
(PDF)

**S1 Data.**
(ZIP)

## Acknowledgments

The authors would like to thank the participants who shared their stories, insights and experiences with us.

## Author Contributions

**Conceptualization:** Yandisa Sikweyiya, Sebenzile Nkosi.

**Data curation:** Yandisa Sikweyiya, Sebenzile Nkosi.

**Formal analysis:** Yandisa Sikweyiya, Sebenzile Nkosi, Malose Langa, Don Operario, Mark N. Lurie.

**Funding acquisition:** Yandisa Sikweyiya, Don Operario, Mark N. Lurie.

**Investigation:** Yandisa Sikweyiya, Sebenzile Nkosi.

**Methodology:** Yandisa Sikweyiya, Sebenzile Nkosi.

**Project administration:** Yandisa Sikweyiya, Sebenzile Nkosi.

**Resources:** Yandisa Sikweyiya, Sebenzile Nkosi.

**Supervision:** Yandisa Sikweyiya, Sebenzile Nkosi.

**Validation:** Malose Langa, Don Operario, Mark N. Lurie.

**Writing – original draft:** Yandisa Sikweyiya, Sebenzile Nkosi, Malose Langa, Don Operario, Mark N. Lurie.

**Writing – review & editing:** Yandisa Sikweyiya, Sebenzile Nkosi, Malose Langa, Don Operario, Mark N. Lurie.

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
