## [Decision Letter · Decision Letter 0]

30 Apr 2021

PONE-D- PONE-D-20-24827

“You see this thing is hard… ey, this thing is painful”: The burden of the provider role and construction of masculinities amongst Black male mineworkers in Marikana, South Africa

PLOS ONE

Dear Dr. Sikweyiya,

Thank you for submitting your manuscript to PLOS ONE. After careful consideration, we feel that it has merit but does not fully meet PLOS ONE’s publication criteria as it currently stands. Therefore, we invite you to submit a revised version of the manuscript that addresses the points raised during the review process.

Although this manuscript holds great promise, it must undergo a major revision before it can be resubmitted to PLOS ONE for another review.

We look forward to receiving your revised manuscript.

Kind regards,

Kathleen Ragsdale

Academic Editor

PLOS ONE

2. Please include additional information regarding the interview guide or script used in the study and ensure that you have provided sufficient details that others could replicate the analyses. For instance, if you developed a guide as part of this study and it is not under a copyright more restrictive than CC-BY, please include a copy, in both the original language and English, as Supporting Information.

"This work was supported by the National Institute of Mental Health and the South African Medical

 Research Council (R01 MH106600) as well as NIH Fogarty Grant (D43TW011308-01. The

content is solely the responsibility of the authors and does not necessarily represent the official

views of the National Institutes of Health or the South African Medical Research Council."

Reviewers' comments:

Reviewer's Responses to Questions

**Comments to the Author**

1. Is the manuscript technically sound, and do the data support the conclusions?

Reviewer #1: Partly

Reviewer #2: Yes

2. Has the statistical analysis been performed appropriately and rigorously? 

Reviewer #1: N/A

Reviewer #2: N/A

3. Have the authors made all data underlying the findings in their manuscript fully available?

Reviewer #1: No

Reviewer #2: Yes

4. Is the manuscript presented in an intelligible fashion and written in standard English?

Reviewer #1: Yes

Reviewer #2: Yes

5. Review Comments to the Author

Reviewer #1: The present manuscript addresses a very interesting area of inquiry into masculinities among Black male mineworkers in Marikana, South Africa. The current version of the manuscript, however, needs to be improved. This manuscript:

1. Is well structured and is analytical with comparative analysis conducted.

2. Needs improvement to ensure better reflection and synthesis.

3. Needs to be reviewed by a copyeditor

The introduction is well written. However, there is a need for the authors to see if information/description on the 2012 protest and social unrest is to be included in this manuscript. If yes, then the findings and discussions should reflect the impact of the incident on the mineworkers.

Descriptions in the methods sections could be improved. In-depth interviews with a small sample of respondents would have resulted in findings which should be richer in content than what presented in the findings section. I believe this could be because the data for question number 5-7 (in the attached interview guide) is not presented in this manuscript.

The discussion section summarizes and reflects back the findings. The authors further supported their arguments with past studies. However, this section could benefit more if theories/concepts of masculinities (eg. by Connell or by theorists on African masculinity/gender) were explicitly used to explain (also by means of agreeing or refuting etc.) the patterns in their findings and that from the literature. Overall, the authors need to take a step back and improve their Results and Discussion section.

Specific comments

L154: Any differences observed between the respondents based on their age, relationship status, and/or ethnic group? Findings presented based on men’s intersecting identities would be interesting to see if there are any similarities differences in findings due to these intersectionality.

L175: Why use theoretical sampling technique? The technique is used for theory building in the grounded theory method. Kindly justify if this is the case. Otherwise, purposive sampling technique is fine.

L177: Since this is a sensitive topic, do you want to omit 'lay counselors'. I imagine there may not be many who are working with the mineworkers and they could be potentially identified?

L180/187-188: Need findings and discussion to reflect this.

L199: If codes/themes were derived from the interview guides, then consider not stating that you used an 'inductive' approach.

L202 – 205: Consider revising the statements - What you are doing here is analyzing the transcripts, assigning codes to data and phenomena to express it in the form of concepts.

L207: Are these references of the 'similar published studies' or of publications on qualitative research which suggests to make such comparisons? If it’s the latter, consider making it explicit in that statement.

L222: Consider removing references from past studies in your Results sections. By having a rich description of the findings alone will enable you to compare the findings with past studies and provide an understanding using theories of masculinities. Also, consider only inserting one or two (preferably short) quotes which reflects each themes/sub-themes in the Results section.

L268-274: Remove the question by the interviewer. An example for you to consider “Ey, you are not recognized [if you are not married, built a house and have a family], you know they do not……”

L416-424: Please check the use of quotations and quotations within quotations - according to the format of PLOS One

L564-568: Consider paraphrasing/summarizing the direct quote

L573: Why personal communication? Can this be replaced with any research?

Reviewer #2: See attachment - Peer Review of PONE-D-20-24827 - “You see this thing is hard… ey, this thing is painful”: The burden of the provider role and construction of masculinities amongst Black male mineworkers in Marikana, South Africa”

6. PLOS authors have the option to publish the peer review history of their article (what does this mean?). If published, this will include your full peer review and any attached files.

Reviewer #1: No

Reviewer #2: **Yes: **Kimberly Kelly

---

## [Author Response · Author response to Decision Letter 0]

30 Jun 2021

A response letter to the reviewers has been uploaded.

---

## [Decision Letter · Decision Letter 1]

4 Feb 2022

PONE-D-20-24827R1

“You see this thing is hard… ey, this thing is painful”: The burden of the provider role and construction of masculinities amongst Black male mineworkers in Marikana, South Africa

PLOS ONE

Dear Dr. Sikweyiya,

Thank you for submitting your manuscript to PLOS ONE. After careful consideration, we feel that it has merit but does not fully meet PLOS ONE’s publication criteria as it currently stands. Therefore, we invite you to submit a revised version of the manuscript that addresses the points raised during the review process.

As you will see, the reviewers are positive about the work and the revisions, and minor revisions are requested. 

We look forward to receiving your revised manuscript.

Kind regards,

Vanessa Carels

Staff Editor

PLOS ONE

Journal Requirements:

Reviewers' comments:

Reviewer's Responses to Questions

**Comments to the Author**

1. If the authors have adequately addressed your comments raised in a previous round of review and you feel that this manuscript is now acceptable for publication, you may indicate that here to bypass the “Comments to the Author” section, enter your conflict of interest statement in the “Confidential to Editor” section, and submit your "Accept" recommendation.

Reviewer #1: All comments have been addressed

Reviewer #2: (No Response)

2. Is the manuscript technically sound, and do the data support the conclusions?

Reviewer #1: Yes

Reviewer #2: Yes

3. Has the statistical analysis been performed appropriately and rigorously? 

Reviewer #1: N/A

Reviewer #2: N/A

4. Have the authors made all data underlying the findings in their manuscript fully available?

Reviewer #1: No

Reviewer #2: (No Response)

5. Is the manuscript presented in an intelligible fashion and written in standard English?

Reviewer #1: Yes

Reviewer #2: Yes

6. Review Comments to the Author

Reviewer #1: The authors have revised the manuscript and adequately addressed the comments raised in the previous round of review.

Reviewer #2: The authors have returned a thorough, thoughtful review that fully addresses the reviewer‘s comments. In particular, the impact of the 2012 mine strike is thoroughly woven into the findings and selection of quotations. There is a more thorough overview of both empirical literature and conceptual frameworks, and theory is woven into the findings in a much more comprehensive matter. This in turn helps establish a clear contribution of the work.

There are a few minor revisions that are still needed.

The authors should consistently use ‘mine work’ or ‘minework.’ The spelling varies throughout the paper.

The word mistress appears on page 13. Please change it to non-marital partner or non-marital sexual partner, as the authors have done throughout the rest of the paper.

The order in which the authors indicate they will discuss the findings are not in the same order in which the findings were actually discussed. Specifically, the second and third themes are switched around.

The sections on provision as an escapable burden and hopes for alternatives for family are very short. In fact they are really too short to fully make the authors’ points. In contrast, the discussion is four pages, The discussion should summarize the findings and explain the significance of the reader. Actual findings should appear above the discussion.

With these changes, I believe the paper will be close to ready for publication.

7. PLOS authors have the option to publish the peer review history of their article (what does this mean?). If published, this will include your full peer review and any attached files.

Reviewer #1: **Yes: **Surendran Rajaratnam

Reviewer #2: No

---

## [Author Response · Author response to Decision Letter 1]

24 Feb 2022

February 24, 2022

The Editor,

PloS One

Dear Prof Vanessa Carels,

Re: Response to reviewers’ comments on manuscript submission: “You see this thing is hard… ey, this thing is painful”: The burden of the provider role and construction of masculinities amongst Black male mineworkers in Marikana, South Africa. 

We thank the editor and reviewers for taking time to review our manuscript. Below, we provide our responses to the reviewers’ comments and detail how we have addressed each comment raised by the reviewers:

Reviewer #1: The authors have revised the manuscript and adequately addressed the comments raised in the previous round of review.

Response

Thank you. 

Reviewer #2: The authors have returned a thorough, thoughtful review that fully addresses the reviewer‘s comments. In particular, the impact of the 2012 mine strike is thoroughly woven into the findings and selection of quotations. There is a more thorough overview of both empirical literature and conceptual frameworks, and theory is woven into the findings in a much more comprehensive matter. This in turn helps establish a clear contribution of the work.

Response

Thank you. 

There are a few minor revisions that are still needed.

The authors should consistently use ‘mine work’ or ‘minework.’ The spelling varies throughout the paper.

Response: 

We thank the reviewer for pointing this out. We have now consistently used ‘minework’ throughout the manuscript. 

The word mistress appears on page 13. Please change it to non-marital partner or non-marital sexual partner, as the authors have done throughout the rest of the paper.

Response: 

The word mistress has been replaced with non-marital sexual partner throughout the manuscript.

The order in which the authors indicate they will discuss the findings are not in the same order in which the findings were actually discussed. Specifically, the second and third themes are switched around.

Response: 

We thank the reviewer for pointing this out. We have now reorganized our discussion in the order in which we proposed to discuss them. 

The sections on provision as an escapable burden and hopes for alternatives for family are very short. In fact, they are really too short to fully make the authors’ points. In contrast, the discussion is four pages, The discussion should summarize the findings and explain the significance of the reader. Actual findings should appear above the discussion.

Response:

We are grateful to the reviewer for this comment and suggestion. We have now expanded the sections on’ Provision as an inescapable demand’ and ‘Creating alternatives’. In both sections we now start by summarizing the key findings, discuss them in relation to relevant literature and then explain the significance of the findings and their implications for gender-transformative interventions. 

With these changes, I believe the paper will be close to ready for publication.

Response

Thanks to the reviewer. We also hope that with the changes we have made in the manuscript, the manuscript will be found to be suitable for publication in PloS One. 

Thank you.

---

## [Decision Letter · Decision Letter 2]

26 Apr 2022

“You see this thing is hard… ey, this thing is painful”: The burden of the provider role and construction of masculinities amongst Black male mineworkers in Marikana, South Africa

PONE-D-20-24827R2

Dear Dr. Sikweyiya,

We’re pleased to inform you that your manuscript has been judged scientifically suitable for publication and will be formally accepted for publication once it meets all outstanding technical requirements.

Kind regards,

Miquel Vall-llosera Camps

Senior Editor

PLOS ONE

Reviewers' comments:

Reviewer's Responses to Questions

**Comments to the Author**

1. If the authors have adequately addressed your comments raised in a previous round of review and you feel that this manuscript is now acceptable for publication, you may indicate that here to bypass the “Comments to the Author” section, enter your conflict of interest statement in the “Confidential to Editor” section, and submit your "Accept" recommendation.

Reviewer #1: All comments have been addressed

2. Is the manuscript technically sound, and do the data support the conclusions?

Reviewer #1: Yes

3. Has the statistical analysis been performed appropriately and rigorously? 

Reviewer #1: N/A

4. Have the authors made all data underlying the findings in their manuscript fully available?

Reviewer #1: No

5. Is the manuscript presented in an intelligible fashion and written in standard English?

Reviewer #1: Yes

6. Review Comments to the Author

Reviewer #1: The authors have adequately revised their article based on the comments provided in the past reviews. The article can now be considered for publication by the editor.

7. PLOS authors have the option to publish the peer review history of their article (what does this mean?). If published, this will include your full peer review and any attached files.

Reviewer #1: **Yes: **Surendran Rajaratnam

---

## [Editor Report · Acceptance letter]

12 May 2022

PONE-D-20-24827R2 

“You see this thing is hard… ey, this thing is painful”: The burden of the provider role and construction of masculinities amongst Black male mineworkers in Marikana, South Africa 

Dear Dr. Sikweyiya:

I'm pleased to inform you that your manuscript has been deemed suitable for publication in PLOS ONE. Congratulations! Your manuscript is now with our production department. 

Kind regards, 

on behalf of

Dr. PLOS Manuscript Reassignment 

Staff Editor

PLOS ONE